# Diabetes Mellitus Should Be Considered While Analysing Sarcopenia-Related Biomarkers

**DOI:** 10.3390/jcm13041107

**Published:** 2024-02-15

**Authors:** Justyna Rentflejsz, Zyta Beata Wojszel

**Affiliations:** 1Doctoral School, Medical University of Bialystok, 15-089 Bialystok, Poland; 2Department of Geriatrics, Medical University of Bialystok, 15-471 Bialystok, Poland; zyta.wojszel@umb.edu.pl

**Keywords:** sarcopenia, type 2 diabetes mellitus, biomarkers, inflammatory markers, myokines, adipokines, hormones, aging, older people

## Abstract

Sarcopenia is a chronic, progressive skeletal muscle disease characterised by low muscle strength and quantity or quality, leading to low physical performance. Patients with type 2 diabetes mellitus (T2DM) are more at risk of sarcopenia than euglycemic individuals. Because of several shared pathways between the two diseases, sarcopenia is also a risk factor for developing T2DM in older patients. Various biomarkers are under investigation as potentially valuable for sarcopenia diagnosis and treatment monitoring. Biomarkers related to sarcopenia can be divided into markers evaluating musculoskeletal status (biomarkers specific to muscle mass, markers of the neuromuscular junction, or myokines) and markers assuming causal factors (adipokines, hormones, and inflammatory markers). This paper reviews the current knowledge about how diabetes and T2DM complications affect potential sarcopenia biomarker concentrations. This review includes markers recently proposed by the expert group of the European Society for the Clinical and Economic Aspects of Osteoporosis, Osteoarthritis and Musculoskeletal Diseases (ESCEO) as those that may currently be useful in phase II and III clinical trials of sarcopenia: myostatin (MSTN); follistatin (FST); irisin; brain-derived neurotrophic factor (BDNF); procollagen type III N-terminal peptide (PIIINP; P3NP); sarcopenia index (serum creatinine to serum cystatin C ratio); adiponectin; leptin; insulin-like growth factor-1 (IGF-1); dehydroepiandrosterone sulphate (DHEAS); C-reactive protein (CRP); interleukin-6 (IL-6), and tumor necrosis factor α (TNF-α). A better understanding of factors influencing these biomarkers’ levels, including diabetes and diabetic complications, may lead to designing future studies and implementing results in clinical practice.

## 1. Sarcopenia: Definition and Epidemiology

Sarcopenia is a chronic, progressive skeletal muscle disease characterised by low muscle strength and mass and worse muscle quality, leading to low physical performance—a hallmark of severe sarcopenia [1]. Its presence significantly worsens prognosis and is related to a higher risk of polypharmacy [2], impaired quality of life [3], falls and fractures [4], loss of physical independence [5], and death [6]. It is a common disorder affecting approximately 10% to 27% of the population aged ≥60 years, depending on the criteria and methods used for diagnosis. The prevalence of severe sarcopenia—a situation in which low muscle mass and strength are accompanied by impaired physical fitness—ranges from 2% to 9% [7]. The frequency of sarcopenia in nursing homes is higher than in community-dwelling older adults and is estimated at 51% in men and 31% in women [8].

Over the years, many well-established scientific groups have proposed different definitions and diagnostic criteria for sarcopenia in search of a tool with supreme sensitivity [9]. Five major research groups have been working to provide a precise description: the European Working Group on Sarcopenia in Older People (EWGSOP), the Asia Working Group for Sarcopenia (AWGS), the International Working Group on Sarcopenia (IWGS), Society for Sarcopenia Cachexia and Wasting Disorders (SCWD), and the Foundation for the National Institutes of Health (FNIH) Sarcopenia Project [1,10,11,12,13,14,15]. Because of that, different cut-off points and definitions applied in the diagnosis of sarcopenia, established over time, may affect the results of previously conducted research.

Currently, the diagnosis of sarcopenia is based on a clinical algorithm proposed by the EWGSOP2 Group in its very latest consensus on sarcopenia. Screening assessment involves using the SARC-F scale. The next step is measuring hand grip strength (or performing a Chair Stand Test) to diagnose low muscle strength, followed by estimating muscle quantity with dual-energy X-ray absorptiometry (DXA), bioelectrical impedance analysis (BIA), magnetic resonance imaging (MRI), or computed tomography (CT). In addition, physical performance tests, such as the Timed-Up-and-Go test (TUG), Short Physical Performance Battery (SPPB), gait speed measurement, and 400 m Walk Test are used to evaluate patients for severe sarcopenia [1]. 

Apart from the above-mentioned anatomic biomarkers of sarcopenia, several specific biochemical biomarkers have been proposed, to be used first of all in research and clinical trials of drugs developed for sarcopenia treatment [16]. They can help to increase knowledge about the mode of action of medications, monitor their effectiveness and side effects, and establish surrogate endpoints [17].

## 2. Type 2 Diabetes and Sarcopenia

Type 2 diabetes mellitus (T2DM) is a metabolic disorder characterised by hyperglycemia determined by insulin resistance and impaired insulin secretion. Patients with T2DM are at higher risk of sarcopenia than euglycemic individuals [18,19,20,21,22,23,24], especially when diabetes complications develop. Diabetic peripheral neuropathy is considered an independent risk factor for sarcopenia in T2DM patients [25]. Similarly, albuminuria is related to sarcopenia in older subjects with T2DM in a dose-related manner, and the risk starts even from minimal albuminuria [26,27]. In a recent meta-analysis by Ai and co-workers, the prevalence of sarcopenia among T2DM subjects was estimated at 18% (95% CI 15–22%), and older age, osteoporosis, and chronic hyperglycemia (measured with HbA1c levels) were significant risk factors. At the same time, lower BMI and metformin administration were protective elements [28]. 

Generally, poor glycemic control in patients with T2DM was linked to low skeletal muscle mass [29]. Some studies even reported it as the leading risk factor for sarcopenia in diabetes, independently of diabetes type or insulin treatment [30]. Still, the effects of anti-diabetic medications on sarcopenia are unclear and need further studies [31,32]. Amongst T2DM patients, those with incident sarcopenia have worse islet function, insufficient insulin secretion, and more significant blood glucose fluctuations than non-sarcopenic patients [33,34,35,36]. In addition, higher body fat content was associated with sarcopenia in older adults with T2DM [37].

Researchers suggest that T2DM and sarcopenia might have a bidirectional causal influence on each other [38,39,40], so sarcopenia can also be a risk factor for developing T2DM in older populations [41]. Furthermore, a study by Huang et al. found 15 common genes and several shared pathways between two diseases [42]. Insulin resistance, mitochondrial dysfunction, advanced glycation end-products (AGEs) and glycogen metabolism, oxidative stress, chronic systemic inflammation, and many other mechanisms play essential roles in the pathogenesis of both sarcopenia and T2DM [43,44]. Sarcopenia might also affect T2DM through decreased production of myokines [45]. Myokines are peptides produced, expressed, and released by muscle fibers that elicit autocrine, paracrine, or endocrine effects [46]. Myokines facilitate communication between muscles and other organs and, among others, can affect lipid and glucose metabolism. Thanks to some myokines’ regulative function, physical exercise may improve insulin sensitivity, but it still needs to be established whether training can enhance insulin secretion [47].

## 3. Biomarkers of Sarcopenia

The early diagnosis of sarcopenia is essential for the success of the therapeutic intervention. However, the diagnostic process is complex and requires specialised equipment. Therefore, the search for alternative diagnostic instruments continues, with attempts to implement biomarker concentrations in advancing the diagnostic process [48,49,50,51,52]. Furthermore, in clinical trials assessing medications to treat sarcopenia, measuring specific biomarker concentrations could help monitor early therapeutic response [16]. Unfortunately, the pathogenesis of sarcopenia includes various complex molecular mechanisms and is not yet fully understood [53]. Therefore, according to the EWGSOP2 group, it is unlikely that one single biomarker could be specific enough to diagnose it, so there is a need to develop a panel of biomarkers instead that could be used for scientific and clinical purposes [1]. 

According to Atkinson and coworkers, a biomarker is “a characteristic that is objectively measured and evaluated as an indicator of normal biologic processes, pathogenic processes, or pharmacologic responses to a therapeutic intervention” [54]. Biomarkers related to sarcopenia can be divided into two groups: Markers evaluating musculoskeletal status (those specific to muscle mass, markers of the neuromuscular junction, muscle turnover, and myokines);Markers evaluating causal factors for sarcopenia (adipokines, hormones, and inflammatory parameters) [16].

Unfortunately, sarcopenia patients can suffer from many comorbidities that may simultaneously affect biomarker concentrations. As it turns out, one of the critical influencing factors that should be considered is T2DM. 

The paper aims to present how diabetes and its complications can potentially affect sarcopenia-related biochemical marker concentrations. This short review concentrates on biomarkers that the expert group of the European Society for the Clinical and Economic Aspects of Osteoporosis, Osteoarthritis and Musculoskeletal Diseases (ESCEO) has suggested recently as those that may currently be useful in phase II and III clinical trials of sarcopenia [16]. The main findings are summarised in Table 1.

### 3.1. Diabetes and Myokines

#### 3.1.1. Myostatin

Myostatin (MSTN), or growth differentiation factor-8 (GDF-8), is a negative regulatory protein of muscle growth. Most studies show that higher myostatin levels are associated with higher muscle mass or function. Yet research results are inconsistent; some studies only found a male-specific association (or no association). Different metabolic profiles and comorbidities might be the reason for these conflicting data results [55]. Studies suggest that MSTN may promote the progression of obesity and diabetes through its effects on skeletal muscle and other non-muscle tissues (adipose tissue, liver) [106]. GDF-8 influences lipid metabolism and can promote intracellular lipid accumulation in muscles by upregulating the expressions of some genes (CD36, PPARγ) [107].

In animal models, diabetic mice’s bones revealed an overexpression of myostatin [108]. Further animal studies showed that decreased myostatin expression improved glucose uptake even in diabetes [109,110]. Moreover, myostatin inhibition caused a reduction in blood glucose, insulin, and triglyceride levels and improved insulin sensitivity [111]. 

In human studies, mean plasma myostatin levels were significantly higher in patients with T2DM [56,57] and subjects with metabolic syndrome and impaired glucose–insulin homeostasis with lower muscle mass [112]. However, some studies showed that T2DM subjects had significantly lower myostatin concentrations, and MSTN was negatively related to blood fasting glucose and triglyceride levels [113]. The weakness of the studies mentioned above is a limited number of participants (ranging from 44 to 128 subjects only). Further studies assessing myostatin levels with larger numbers of T2DM participants need to be performed in order to establish the biomarker’s significance in this specific group.

A study by Li et al. indicated that among patients with T2DM, myostatin levels were significantly higher in subjects with sarcopenia [114]; however, the study only included 34 sarcopenic individuals (meeting the criteria of the Asian sarcopenia diagnostic consensus). Exercise programs decrease MSTN concentrations in T2DM patients [57,115]. Myostatin concentration levels were also positively associated with diabetic retinopathy [116]. Higher levels of muscle myostatin mRNA content were found in T2DM patients. However, plasma myostatin concentrations did not differ between subjects with diabetes and controls [117]. 

Children with type 1 diabetes mellitus (T1DM) exhibited significantly elevated serum myostatin levels compared to healthy subjects in some studies [118,119]. On the other hand, gestational diabetes does not influence cord blood myostatin levels [120]; however, it affects myostatin protein expression in placentae [121].

#### 3.1.2. Follistatin

Follistatin (FST) acts as an antagonist of transforming growth factor-β (TGF-β) ligands (including myostatin and activin A), which is why MSTN and FST should be considered together. Follistatin concentrations seemed to be elevated in patients with sarcopenia, as was indicated in a large cross-sectional study of community-dwelling postmenopausal women [58], yet some studies showed no correlation between this myokine and muscle health [122,123,124]. Still, it is worth mentioning that in one of these studies, follistatin levels tended to be negatively correlated with muscle mass, with borderline statistical significance, and significantly correlated in female patients [122], and the other two mentioned studies had relatively small sample sizes.

In animal models, intravascular FST gene administration to prediabetic and diabetic mice improved glycaemic control and promoted compensatory insulinemia [125]. Human studies indicated plasma FST levels increase in T2DM patients [59,60]. In an observational study by Wu et al., subjects who developed T2DM during follow-up had higher FST concentrations at baseline, suggesting that increased FST levels were associated with a higher risk of diabetes [126]. In T2DM patients, exercising was connected with increasing follistatin levels [115]; nonetheless, exercise-induced FST secretion seemed to be impaired in patients with T2DM [127].

#### 3.1.3. Irisin

Irisin is a myokine that induces muscle hypertrophy. Its concentrations were found to be decreased in sarcopenic patients [61,62,63,64,65,66], but some studies found no correlation between the two [128,129]. There are multiple studies proving irisin levels are significantly lower in patients with T2DM (including a meta-analysis published in 2016 and involving 1745 diabetic patients and 1337 controls from 17 cross-sectional and 6 case–control studies [67] and another published in 2021 [68] and evaluating 26 studies with a total of 3667 participants), yet some studies show the opposite correlation [113,130]. Nevertheless, we must underline that studies with the opposite results involved much fewer participants (73 diabetic individuals with 55 controls and 104 T2DM subjects with 124 controls, respectively).

Decreased irisin concentrations in T2DM patients are observed in parallel with cardiovascular complications [131,132,133]. They may have some predictive value in heart failure diagnosis [134,135] and be linked to nephropathy [136]. Some interventions, like sitagliptin treatment, increase serum irisin levels in T2DM patients [137].

### 3.2. Diabetes and Markers of the Neuromuscular Junction and Neuroinflammation

Brain-derived neurotrophic factor (BDNF) and glial cell line-derived neurotrophic factor (GDNF) are markers associated with the neuromuscular junction and neuroinflammation. These are neurotrophins released from both neurons and muscles and play an essential role in muscle development and metabolism and the regulation of synapse function [138]. Research on these neurotrophins’ levels in sarcopenia is limited. In a Japanese study of haemodialysis patients, BDNF concentrations were lower in subjects with severe sarcopenia and frailty [69]. Another study showed an association of lower BDNF levels with a low skeletal muscle index (SMI) in kidney transplant recipients [70]. In other studies, plasma concentrations of BDNF were higher in more advanced stages of sarcopenia [139]. Both BDNF and GDNF were shown to potentially play a role in the diagnosis of sarcopenia in chronic obstructive pulmonary disease (COPD) and Parkinson’s disease (PD) patients [140,141]. A recent meta-analysis of 28 studies, including 2734 diabetic individuals and over 6000 controls, revealed significantly decreased BDNF concentrations in T2DM subjects and patients with diabetic retinopathy compared to controls [71].

### 3.3. Diabetes and Procollagen Type III N-Terminal Peptide

Procollagen type III N-terminal peptide (PIIINP; P3NP) is produced during type III collagen synthesis. Its concentrations seem to be related to aging, body composition, and physical performance [142,143]. The evidence of its role in sarcopenia is yet to be established. One study showed an association between higher PIIINP levels and lower lean body mass in postmenopausal women [72], yet in another study, PIIINP concentrations were associated with an increased SMI in men [73]. These discrepancies might be related to steroid hormones levels, as some studies found that testosterone may increase plasma PIIINP levels in a dose-dependent manner [144] and higher PIIINP concentrations induced by testosterone treatment were related to greater gains in skeletal muscle mass in older males [145].

The PIIINP levels increase with higher BMI and age [146], but there seems to be no significant correlation between them and diabetes [74]. On the other hand, they might be associated with diabetes-related complications such as proliferative retinopathy [147] and peripheral vascular disease [148].

### 3.4. Diabetes and Sarcopenia Index

Sarcopenia index (SI; serum creatinine to serum cystatin C ratio) is a recently developed formula for evaluating muscle mass [149]. Serum creatinine concentrations reflect muscle protein turnover but are also determined by renal function. That is why, in this formula, creatinine level is related to another marker of renal function—cystatin C. 

SI has been correlated with CT skeletal muscle cross-sectional surface area [149,150,151], calf circumference, BMI, handgrip strength [152], and gait speed [153]. In some studies, SI was found not to be accurate enough to diagnose sarcopenia in community-dwelling older individuals [154,155], but there was evidence of its accuracy in hospitalised older patients [75,76]. 

Some studies suggested that in T2DM patients, SI could be a valuable screening sarcopenia marker [156,157]. It was also related to subclinical atherosclerosis [158] and increased risk of fractures [159] in T2DM subjects, as well as to osteoporosis in T2DM male patients [160]. A low serum creatinine-to-cystatin C ratio was associated with a higher risk of all-cause mortality in diabetic patients [161]. A higher SI normalised by body weight was associated with a reduced risk of T2DM in middle-aged and older subjects [77].

### 3.5. Diabetes and Adipokines

Adiponectin is an adipokine with insulin-sensitising and anti-inflammatory effects. Reducing body fat leads to increased adiponectin blood concentrations [162]. Although several research results were conflicting, a recent meta-analysis (involving six cross-sectional and one retrospective study with a total number of 1389 participants) confirmed that circulating adiponectin levels were higher in sarcopenic patients, and the female sex significantly influenced the results. These greater plasma adiponectin concentrations, independent of BMI or fat mass, might be explained partially by sex hormones. It is discussed that adiponectin could play a role in a compensatory mechanism for mitigating sarcopenia resulting from chronic inflammation and oxidative stress [78]. Lower adiponectin levels were associated with T2DM and metabolic syndrome [79]. The decrease in adiponectinemia was a risk factor in the pathogenesis of insulin resistance, T2DM, and cardiovascular pathologies [162]. In in vitro studies on skeletal muscle cells, adiponectin increased glucose uptake by translocating the glucose transporter GLUT4 to the cell surface [163]. The activation of muscle protein degradation via the ubiquitin–proteasome proteolytic pathway mediated by insulin resistance may result from reduced adiponectin levels [164].

Leptin is an adipokine that, among other things, improves immune response and induces lipid catabolism [165]. Some studies showed leptin levels were elevated in sarcopenic patients [80,81]. Nevertheless, research in this area yields conflicting data. A study by Lin et al. indicated that lower serum leptin concentrations were independently associated with sarcopenia in haemodialysis patients [82]. The weakness of these studies is a limited number of participants. Another study of community-dwelling older adults showed a positive association of leptin serum levels with muscle and fat mass and negative with muscle strength—increased leptin levels were linked to a higher risk of dynapenia (low muscle strength), but a lower risk of sarcopenia [83]. In addition, patients with sarcopenic obesity have higher leptin and lower adiponectin levels compared to sarcopenic non-obese subjects [166]. Leptin concentrations in sarcopenic individuals, specifically in patients with sarcopenic obesity and in sarcopenic T2DM patients, need further studies. 

Leptin levels seem to be increased in individuals with obesity [167] as well as in diabetic patients [84,85,86,87]. Yet, some studies found such an association only in obese T2DM patients (in a study by Liu et al., leptin concentrations were significantly lower in newly diagnosed T2DM patients with normal BMI than healthy individuals but considerably higher in obese T2DM patients) [168]. Elevated leptin concentrations may be related to insulin resistance in diabetic patients [169,170,171]. Leptin was also higher in T2DM patients with diabetic complications [172,173]. Although some studies suggested a difference in leptin concentrations between men and women [174,175], a recent meta-analysis of 10 studies with more than thirty thousand diabetic participants in total concluded gender factors did not affect leptin levels in T2DM patients [176]. In addition, many other factors may influence leptin levels in T2DM patients, for example, some anti-diabetic medications [177,178].

### 3.6. Diabetes and Hormones

Insulin-like growth factor (IGF-1) regulates muscle anabolic and catabolic pathways, inhibiting muscle atrophy and potentiating muscle regeneration [179]. IGF-1 concentrations seem to be higher in younger subjects [123], and the age-related decline in IGF-1 levels is considered a possible factor contributing to the development of sarcopenia [180]. There is evidence of decreased levels of IGF-1 in sarcopenic patients [88,89,90,91,92], yet a study by Jiang and co-workers found such an association only in men [181]. 

Decreased IGF-1 concentrations were also reported in patients with T2DM [93,94]. However, some studies found no association between IGF-1 levels and incident diabetes in the general population [95], and studies that associated low IGF-1 levels with diabetes in young adults did not confirm such an association in subjects over 65 years of age [96]. In contrast, a survey by Shao and colleagues found significantly higher IGF-1 levels in T2DM patients compared to the control group, and IGF-1 concentrations increased with a more significant level of albuminuria [97]. This might suggest that different comorbidities and diabetic complications can alter IGF-1 concentrations. Elevated levels of IGF-1 were found in older patients with dementia and diabetes [182], yet in another study, lower IGF-1 levels were a predictor of poor cognitive performance in patients with diabetes [183]. In T2DM subjects with relatively reasonable glycaemic control, increased IGF-1 was related to a higher risk of retinopathy progression [184]. On the other hand, another study of T2DM patients revealed lower IGF-1 concentrations in subjects with T2DM complicated by coronary heart disease [185].

Dehydroepiandrosterone sulphate (DHEAS) is a steroid hormone, a testosterone precursor. Its role in sarcopenia has long been studied, and the age-related decrease in its levels is perceived as a causal factor for muscle loss. DHEAS concentrations correlate positively with older patients’ skeletal muscle mass and strength and seem to be decreased in sarcopenic individuals [58,98,99]. Exercise and nutritional supplementation can increase DHEAS levels in older adults, and this effect is more pronounced in frail participants [186]. Among T2DM patients, those with sarcopenia have significantly decreased DHEAS levels [187].

Study results suggest that serum DHEAS concentrations are negatively associated with the risk of T2DM [100,101], although some studies indicate that DHEAS levels have no association with incident diabetes in women [101,188]. In a study with a 5-year follow-up, the decline in DHEAS was especially significant in subjects who became diabetic [102]; another study revealed that higher DHEAS concentrations at baseline were a protective factor against developing T2DM in older men (but not in older women) [189]. Lower DHEAS levels were associated with poor diabetes control [190]. In addition, DHEAS levels correlated inversely with the degree of urinary albumin excretion in T2DM male patients [191] and with the duration of diabetes [192]. Decreased DHEAS concentrations in T2DM patients were also associated with diabetic retinopathy [193], atherosclerosis [194,195], diabetic kidney disease [196], and coronary heart disease in diabetic men [197].

### 3.7. Diabetes and Inflammatory Markers

One of the crucial mechanisms that may be related to sarcopenia pathogenesis is chronic inflammation. Inflammatory cytokines have been linked to muscle wasting, promoting protein catabolism and suppressing muscle tissue synthesis. A meta-analysis published in 2016 indicated that sarcopenia is associated with increased C-reactive protein (CRP) levels but did not confirm the association between sarcopenia and interleukin-6 (IL-6) or tumour necrosis factor α (TNF-α) concentrations [103]. Nevertheless, a more recent meta-analysis with the significantly greater number of articles analysed, concluded that increased CRP, IL-6, and TNF-α levels were significantly associated with lower skeletal muscle strength and muscle mass [198].

Elevated levels of inflammatory markers were also linked to an increased risk of T2DM, as confirmed in large meta-analyses [104,105]. Additionally, T2DM patients with depression or cognitive impairment seem to have elevated IL-6 and CRP levels compared to diabetic subjects without comorbidities [199,200]. However, several interventions in T2DM patients, such as diabetes medications or exercise training, can decrease inflammatory biomarker concentrations. Metformin administration reduces CRP (but not TNFα or IL-6) levels [201]. Exercise decreases CRP, TNFα, and IL-6 in diabetic patients [202,203,204]. 

## 4. Conclusions

Sarcopenia and type 2 diabetes are chronic diseases that affect many older people. Their pathogeneses are interconnected, and their impact is bidirectional—each is a risk factor for the other. Both diseases share several common determinant genes and common pathogenetic pathways. The assessment of serum concentrations of various biomarkers may provide new information about these pathomechanisms and help develop new tools for rapid diagnosis and monitoring sarcopenia treatment. 

This brief review shows that using a single biomarker concentration to diagnose or monitor sarcopenia would be insufficient because of different factors influencing its levels. In some cases, the results of available studies are inconsistent and potential reasons for discrepancies include variations in study design, differences in sample sizes, or distinct characteristics of the study populations. What is more, most of the biomarkers presented require further studies to assess their levels in patients with various comorbidities. These include, among others, diabetes status, the level of glycaemic control, diabetes complications, and glucose-lowering medications. Therefore, diabetes mellitus should be considered an important confounding factor in future analyses of studies designed to assess levels of sarcopenia-related biomarkers. 

Further studies should be designed to develop a set of biomarkers that would be suitable for monitoring sarcopenia and its treatment in T2DM patients. This may result in a better understanding of the factors influencing the concentration of various biomarkers in serum and establish their usefulness in clinical practice. 

## Figures and Tables

**Table 1 jcm-13-01107-t001:** Changes in biomarker concentrations in sarcopenia and in T2DM (based on [16]).

Biomarker	Modification in Sarcopenia	Modification in T2DM	Mechanism of Action/Comments
Assessment of musculoskeletal status			
Myokines			
MSTN (GDF-8)	↓ conflicting sex-dependent data [55]	↑ [56,57]	Mechanism: muscle protein turnover Increases with higher muscle mass LC-MS/MS measurement methods should be favored
FST	↑ [58]	↑ [59,60]	Mechanism: muscle protein turnover Suggested to be assessed together with MSTN MSTN and FST are acutely increased after exercise but return to baseline within 24 h
Irisin	↓ [61,62,63,64,65,66]	↓ [67,68]	Mechanism: muscle protein turnover
Neuromuscular junction			
BDNF	↓ [69,70]	↓ [71]	Mechanism: remodelling of neuromuscular junction
Other			
PIIINP	limited number of studies, conflicting data between sex groups [72,73]	probably no correlation [74]	Mechanism: muscle collagen turnover
SI	↓ [75,76]	probably ↓ a limited number of studies [77]	Developed to reflect muscle mass, reclassified as the marker of muscle turnover subclass; = serum creatinine (mg/dL)/serum cystatin C (mg/L)) × 100
Assessment of causal factors			
Adipokines			
Adiponectin	↑ [78]	↓ [79]	Anti-inflammatory factor Mechanism: muscle–fat crosstalk
Leptin	limited number of studies, conflicting data [80,81,82,83]	↑ [84,85,86,87]	Relevant in sarcopenic obesity
Hormones			
IGF-1	↓ [88,89,90,91,92]	↓ [93,94] conflicting data [95,96,97]	Age-related differences Anabolic
DHEAS	↓ [58,98,99]	↓ [100,101,102]	Sex-dependent differences Anabolic
Inflammatory markers			
CRP	↑ [103]	↑ [104,105]	Mechanism: low-grade chronic inflammation Preferred use of ultrasensitive method
IL-6	probably no basal modification [103]	↑ [104,105]	Mechanism: low-grade chronic inflammation Increases associated with lower muscle strength and mass
TNF-α	probably no basal modification [103]	↑ [105]	Mechanism: low-grade chronic inflammation Increases associated with lower muscle strength and mass

Where: ↑, increased concentration; ↓, decreased concentration; BDNF, brain-derived neurotrophic factor; CRP, C-reactive protein; DHEAS, dehydroepiandrosterone sulphate; FST, follistatin; GDF-8, growth differentiation factor-8; GDNF, glial cell line-derived neurotrophic factor; IGF-1, insulin-like growth factor; IL-6, interleukin-6; LC-MS/MS, liquid chromatography–tandem mass spectrometry; MSTN, myostatin; PIIINP, procollagen type III N-terminal peptide; SI, Sarcopenia Index; T2DM, type 2 diabetes mellitus.

## Data Availability

Not applicable.

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
