# Peer review of "Diabetes Mellitus Should Be Considered While Analysing Sarcopenia-Related Biomarkers"

_jcm, 2024, doi:10.3390/jcm13041107_

Round 1

Reviewer 1 Report

Comments and Suggestions for Authors

Dear authors,

I have studied with great interest the manuscript “Diabetes mellitus should be considered while analysing sarcopenia-related biomarkers”.

This article provides a comprehensive review of the intricate relationship between sarcopenia and type 2 diabetes mellitus (T2DM). It critically examines the current understanding of how diabetes and its complications influence potential biomarker concentrations associated with sarcopenia. The review underscores the importance of considering diabetes and its complications to better interpret these biomarker levels, offering insights for the design of future studies and the application of results in clinical practice.

But I have some comments to improve the quality of the presentation.

1.The term "frailty" appears in the keywords but is not extensively discussed in the manuscript. It is advised to reconsider its inclusion in the keywords, otherwise its deletion should be considered.

2.Sections 3.5. and 3.6. appear to be missing, and it is suggested that they be included to make the presentation more complete;

3.Regarding conflicting findings, it is hoped that the authors will provide additional insights into potential reasons for discrepancies, such as variations in study design, differences in sample sizes, or distinct characteristics of the study populations.

4.For studies mentioned in the discussion, especially those involving meta-analyses, it is suggested to furnish more detailed information about these studies to assess their reliability and consistency.

Reviewer 2 Report

Comments and Suggestions for Authors

This manuscript aims to describe the impart of DM on the expression levels of sarcopenic-related biomarkers. Indeed, some of the well-known biomarkers of sarcopenia might be affected with the present of others comorbidities. This manuscript is well written, and here are some comments from me to improve the manuscript.

1. Please include citations for each biomarkers in table 1. Which reference are you referring to for the expression levels of each biomarkers?

2. Since the authors concluded that the sarcopenic related biomarkers might be affected by the present of DM, the authors may consider to further discuss on the suitability of using such biomarkers for monitoring or detection of sarcopenia among DM patients, or any precautions steps that could be taken when measuring the sarcopenic-related biomarkers levels in DM patients. 
